# Spatiotemporal Changes and the Driving Forces of Sloping Farmland Areas in the Sichuan Region

**Meijia Xiao [1]**, **Qingwen Zhang [1,***, **Liqin Qu [2]**, **Hafiz Athar Hussain [1]**, **Yuequn Dong [1]** **and Li Zheng [1]**

1. Agricultural Clean Watershed Research Group, Institute of Environment and Sustainable Development in Agriculture, Chinese Academy of Agricultural Sciences/Key Laboratory of Agro-Environment, Ministry of Agriculture, Beijing 100081, China; 13121281099@163.com (M.X.); atharhussainswu@yahoo.com (H.A.H.); dongyuequn@163.com (Y.D.); zhengliswu@163.com (L.Z.)
2. State Key Laboratory of Simulation and Regulation of Water Cycle in River Basin, China Institute of Water Resources and Hydropower Research, Beijing 100048, China; liqin.qu@foxmail.com
* Correspondence: zhangqingwen@caas.cn; Tel.: +86-10-82106031

**Abstract:** Sloping farmland is an essential type of the farmland resource in China. In the Sichuan province, livelihood security and social development are particularly sensitive to changes in the sloping farmland, due to the region's large portion of hilly territory and its over-dense population. In this study, we focused on spatiotemporal change of the sloping farmland and its driving forces in the Sichuan province. Sloping farmland areas were extracted from geographic data from digital elevation model (DEM) and land use maps, and the driving forces of the spatiotemporal change were analyzed using a principal component analysis (PCA). The results indicated that, from 2000 to 2015, sloping farmland decreased by 3263 km$^2$ in the Sichuan province. The area of gently sloping farmland ($<10°$) decreased dramatically by 1467 km$^2$, especially in the capital city, Chengdu, and its surrounding areas. However, the steep sloping farmland ($>25°$) decreased by 302 km$^2$, and was still the largest portion of total farmland in the area. The PCA analysis indicated that the main driving forces behind the changes were social and economic factors. The influence of agricultural intensification factors, such as the multiple cropping index and sown areas of crops, was relatively weak. Given the decrease in the overall slope cultivated area and the increased portion of moderately steep slope land ($10–25°$) in the cultivated area, special attention should be paid to the scientific conservation of sloping farmland during rapid social and economic development.

**Keywords:** spatiotemporal change; driving force; sloping farmland area; GIS image

## 1. Introduction

Cultivated land is a fundamental resource for human survival in developing countries [1–3], including China, which has 22% of the world's population and 7% of the world's cultivated land [4,5]. Sloping farmland refers to cultivated land with a slope $>2°$ [6], which is a key component of relatively scare cultivated land resources, accounting for 28.35% of the total domestic cultivated land in China [7,8]. Southwest China is the region with the largest grain self-sufficiency, where the proportion of sloping farmland accounts for 90% of the cultivated land [9]. Sloping farmland is characterized by a serious loss of soil nutrients, and a low and unstable yield [10–13].

Quantitative/qualitative analysis on land use and land use change (LUC) is crucial to support public choices of land use management [14]. With the introduction of geographic information system (GIS) techniques, LUC assessment becomes a fundamental tool to support analysis of the causes and consequences of land use dynamics [15,16]. Previous studies have shown that the area of farmland

has significantly decreased because of degradation and the conversion of farmland into building land [17,18], where the increasing speeds of construction and forest land are faster than other land use types [19].

Slope cropland is an ecosystem involving the interaction of both natural and social factors [20–22]. The evolution of the sloping farmland system has not only been influenced by natural conditions, but also by social and economic factors [23,24]. The decrease in human-induced cultivated land has become a serious problem, particularly with the increasing global population growth and the unreasonable utilization of water and soil resources by human beings [25–28]. Humans' social and economic activities have played a decisive role in changing cultivated land, where the impact of socio-economic and related industrial policies has been felt over a short period of time, including population growth, agricultural structure adjustment, urbanization, GDP per capita, output value, etc. [29,30]. Nowak and Scheifder found that sloping farmland was seriously affected by agricultural activities, and that the degradation areas were mainly located at the agriculturally intensive areas [31]. The International Geosphere-Biosphere Programme (IGBP) and the International Human Dimensions Program on Global Environmental Change (IHDP) have identified the main driving forces of land-use and land-cover, which are natural environments, land-use management, and social and economic factors [32].

The subject of most studies regarding land use change has been the transformation of land areas of different utilization types, such as cultivated land, woodland, garden, and construction lands. However, studies on the spatiotemporal change analysis of a certain land area, such as sloping farmland, are sparse. Some studies have been conducted on the change of cultivated land [33–35]. However, the changes of sloping farmland still need further discussion. There have been some studies on the influence of natural factors on sloping farmland [36–38], but the study of human factors on sloping farmland is still insufficient. Study of the impact of social factors, economic factors, and agricultural intensity level is still needed Moreover, the influence of natural factors on sloping farmland occurs over a large time span and cannot be reflected in a short period, while the human factors, though complex, are more easily controllable [39]. It is necessary to gain a deeper understanding of the basic change rule of sloping farmland, as well as the human driving factors that cause the change, to formulate policies for preventing and controlling the degeneration of sloping farmland.

In this paper, LUC dynamics in Sichuan were examined, with specific attention to the sloping farmland. We took the sloping farmland area extracted by GIS as the research object, and we analyzed the change law of the sloping farmland at different ranges with the lapse of time, and the spatial variation among the Sichuan Province. Additionally, we used statistical techniques to quantify the relationships between land use and driving forces. Principal component analysis and correlation analysis were used to reveal the relationship between slope cropland and social factors, economic factors, and agricultural intensity factors. Based on the change law of cultivated slopes and the characteristics of regional driving force change, we wanted to provide a basis for the sustainable development of sloping farmland.

## 2. Data and Methods

### 2.1. Study Area

The Sichuan region covers an area of $19 \times 10^4$ km$^2$ and is located in the upper reaches of the Yangtze River (Figure 1), where it is surrounded by mountains, and is dominated by 5:3:2 proportions of mountains, highlands, and hilly areas. The province lies in the subtropical humid climate zone, with an annual average temperature ranging from 14 to 19 °C, which is 1 °C higher than the average at equivalent latitudes. The annual cumulative daily average temperature $\geq 10$ °C ranges from 4200 to 6100 °C, and the annual sunshine varies from 900 to 1600 h. Rainfall is abundant with annual precipitation ranging from 900 to 1200 mm. Sichuan province was selected as the typical region because of its large sloping farmland area.

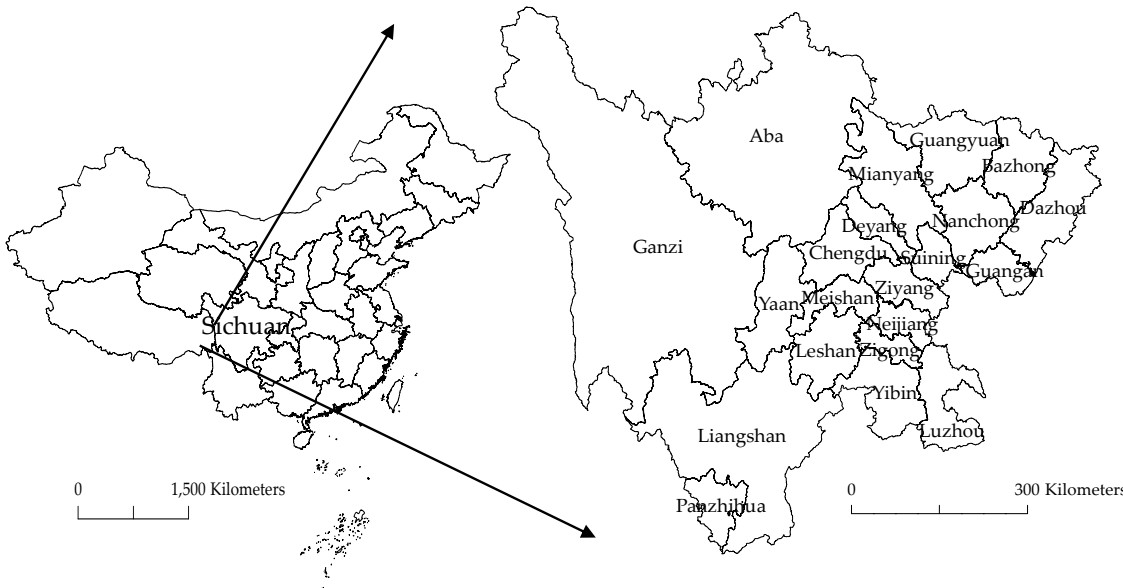

**Figure 1.** Location of the study area. The photo on the left represents the location of the study area, and the right hand photo represents distribution of the cities (states) of Sichuan province.

*2.2. Data Sources*

The input data for this study included the sloping farmland area and the driving force data. Social factors (total population, urbanization rate), economic factors (per capita GDP, output value of the tertiary industry, total mileage of highways, construction area), and the agricultural intensity factors (sown area of crops and the multiple cropping index) were included in the driving force.

2.2.1. Data of Sloping Farmland Area

Taking the grading standard of the secondary survey of land resources as a reference, according to the topography of the research area, the small unit slope was re-extracted to calculate 2–5°, 5–10°, 10–15°, 15–20°, 20–25°, and >25° grades of sloping farmland areas in the Sichuan province using the ArcGIS 10.1 software. The extraction method of the slope cultivated area is shown in Figure 2. We used the administrative division map of Sichuan province as a base map, where the administrative division map was sourced from the Resource and Environmental Science Data Center of the Chinese Academy of Sciences (http://www.resdc.cn). We then combined the administrative division map with the land use maps with a resolution of 30m, which were provided by the Data Center for Resources and Environmental Sciences, Chinese Academy of Sciences (RESDC) (http://www.resdc.cn), for Sichuan province in 2000, 2005, 2010, and 2015, as well as the DEM with a resolution of 15m of Sichuan province, provided by the Geographical Information Monitoring Cloud Platform (http://www.dsac.cn). A 3D analyst–raster surface–slope was used to convert the DEM to slope in the GIS, where the basic principle is shown in Equation (1), as in Reference [40]:

$$S = \arctan\sqrt{f_x{}^2 + f_y{}^2} \tag{1}$$

where $f_x$ refers to the elevation change rate in the south–north direction, and $f_y$ refers to the elevation change rate in the east–west direction.

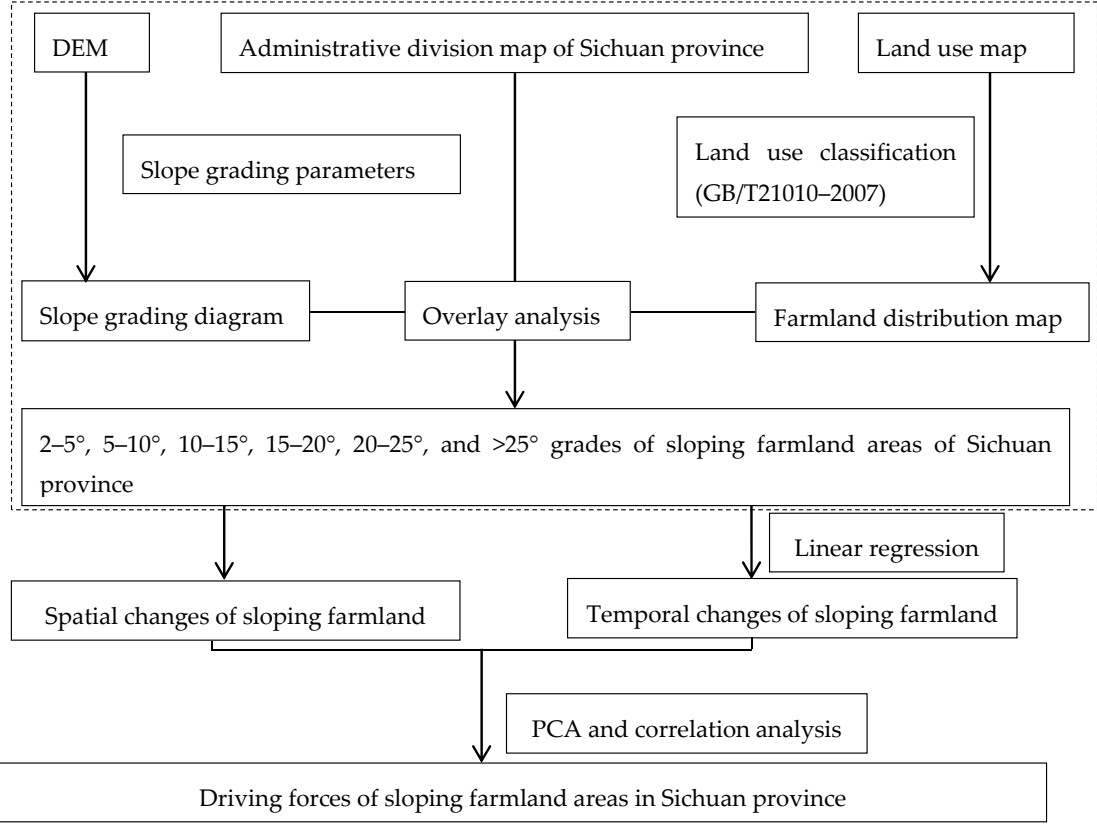

**Figure 2.** The flow chart of spatiotemporal changes and driving forces of sloping farmland areas. Note: Figure dash parts represent extraction method of slope cultivated area by GIS.

### 2.2.2. Data of Driving Forces

Social factors, economic factors, and agricultural intensive factors were derived from the China Statistical Yearbook (http://www.stats.gov.cn/).

### *2.3. Methods*

Land use change analysis relied on geographic data including province and city scale; spatial, temporal, and quantitative analysis were used to measure and study sloping farmland area change. Statistical techniques were used to quantify the relationships between land use and driving forces. We analyzed the spatial and temporal changes of sloping farmland area based on data extracted by GIS, using linear regression and direct analysis. Principal component analysis and correlation analysis were then used to reveal the direct correlation between slope cropland and driving forces, and demonstrate the main driving forces. The flow chart of this article was showed in Figure 2. Data processing and analysis were completed by R 3.4.3 and SPSS18.0 software. The diagrams were made by arcgis10.1 and origin 8.5.

### 2.3.1. Spatiotemporal Changes Analysis of Sloping Farmland

We found the variation rule of sloping farmland based on the extracted data directly. Additionally, linear regression analysis was adopted to predict the change trends of sloping farmland areas using extracted data. We applied unary linear regression at 95% confidence interval as a function of the slope cropland data, in which the time was an independent variable. F-test ($\alpha = 0.05$) was used to test the significance of the unary regression equation [41].

2.3.2. Driving Force Analysis

Principal component analysis (PCA) is a statistical procedure that applies an orthogonal transformation in order to convert a set of observations which might be possibly correlated variables (driving force indexes) into linearly uncorrelated variables called principal components. To get the main driving forces of the change of sloping farmland and the regional differences in the main driving forces, the natural environmental and socio-economic factors affecting the change of sloping farmland, eight indexes were adopted to assess population pressure, social and economic development, urbanization level, agricultural intensification factors, and farmers' life demands, and were analyzed by PCA. Analysis steps of PCA were as follows [42,43]:

1. Establish the original matrix of the dataset. The original data can be represented by driving force indexes matrix (including m indexes and n variables):

$$X_0 = \begin{bmatrix} x_{11} & \cdots & x_{1m} \\ \vdots & \ddots & \vdots \\ x_{n1} & \cdots & x_{nm} \end{bmatrix} \tag{2}$$

2. Data standardization. $X_0$ should be normalized first to eliminate the influence caused by different magnitudes of variables in $X_0$. As we know, driving factors, including social factors, economic factors, and agricultural intensive factors have different units. Non-dimensional treatments for the original data were produced through z-score method, the data z-score normalization is shown in Equation (3) [44]:

$$y_{ij} = \frac{x_{ij} - x_j}{S_j} \tag{3}$$

where $y_{ij}$ represents the standardized value of factor ij and varies from 0 to 1; i = 1,2,3 . . . I, j = 1,2,3 . . . J, $X_{ij}$ was the jth driving force index in the ith year; $X_j$ and $S_j$ were the driving force mean and standard deviation jth index, respectively.

3. Correlation matrix R was calculated using Equation (4)

$$R = \left[ r_{ij} \right]_{n \times m} = \frac{ZZ'}{N - 1} \tag{4}$$

4. Solving J eigenvalues of R: $\lambda 1 \geq \lambda 2 \geq \ldots \geq \lambda J$, and the corresponding eigenvector u1^', u2^', u3^' . . . .un^' were orthonormal, and u1^', u2^', u3^', . . . .un^' were the principal axis.

5. The principal component was calculated through Equation (5):

$$U = \begin{bmatrix} u_1' \\ \vdots \\ u_n' \end{bmatrix} = \begin{bmatrix} u_{11} & \cdots & u_{1m} \\ \vdots & \ddots & \vdots \\ u_{n1} & \cdots & u_{nm} \end{bmatrix} \tag{5}$$

6. Data filter. The maximum load index and the index with the load below 10% of the maximum load were retained in each principal component to reflect the driving force principal factors in the sloping farmland area change in different regions [45,46].

However, PCA cannot explain the direct correlation between driving forces and sloping farmland. In this study, we combined the principal component analysis data with the results of correlation analysis data, and the main driving force types of slope cultivated land change in Sichuan Province were revealed. The correlation between the driving force index and the sloping farmland of Sichuan province was analyzed using the Spearman correlation analysis method.

## 3. Results

### 3.1. Land Use Change in Sichuan Province from 2000 to 2015

The land use types were divided into six categories: farmland, forestland, grassland, water, building land, and unused land [47]. From 2000–2015, the changes in land use types in Sichuan province indicated the decrease of farmland and the rapid increase of building land (Figure 3). In terms of space scale, farmland was reduced from 25.1% to 24.6% of the entire land during the study period. Building land increased to 1754 km$^2$, with an annual average growth of 116.9 km$^2$. The data also showed that the area of buildings accounted for 0.60% of the total area in 2000, and 1.0% in 2015.

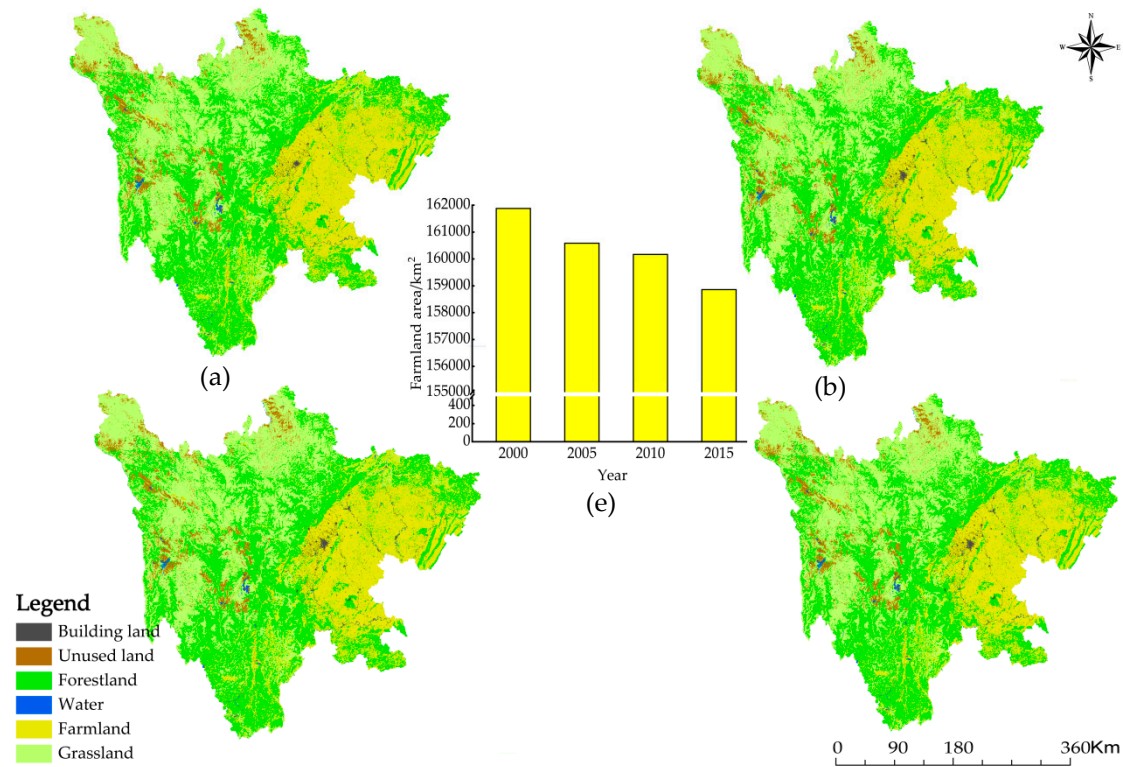

**Figure 3.** Land use type distribution maps of the Sichuan basin in different periods. Note: (**a**–**d**) represent the land use map of Sichuan province in the years 2000, 2005, 2010, and 2015, respectively. Subfigure (**e**) represents the farmland change from 2000 to 2015.

### 3.2. Temporal Variation of Sloping Farmland Area

Sloping farmland, which refers to a gradient degree of more than 2°, accounted for more than 90% of Sichuan province. The proportions of slope cultivated land were 31% for 5–10° sloping farmland, 20% for 10–15°sloping farmland, 18% for 2–5°sloping farmland, 11% for >25°sloping farmland, and 8% for 20–25° sloping farmland. The proportion of gentle sloping farmland (<10°) decreased, while the proportion of moderately sloping farmland (10–25°) increased (Figure 4).

Slope cropland area has been decreasing over the past 15 years in Sichuan province. The decreased area was 1207 km$^2$ from 2000–2005, 400 km$^2$ from 2005–2010, and 1217 km$^2$ from 2010–2015, respectively. Sloping farmland that was less than 10° decreased to 1646.82 km$^2$ from 2000 to 2015; greater than the decline in the 25° sloping farmland which decreased by 148.5, 79.8, and 72.6 km$^2$ from 2000 to 2005, 2005 to 2010, and 2010 to 2015, respectively. Through the unary linear regression analysis, the cultivated land area of 2–5°, 5–10°, 10–15°, 15–20°, 20–25°, and >25° decreased linearly with the increase in the number of years. The coefficients of determination, R$^2$, were 0.92, 0.93, 0.95, 0.97, 0.97, and 0.95, respectively, and the negative regression coefficients were −44.05, −58.1, −28.85, −16.34,

−10.4, and −19.72, respectively. It was implied that the fastest decline in sloping arable land was experienced in the gentle sloping farmland less than 10°, while the slowest decline was experienced in the 20–25° slope cultivated land. On the basis of trend prediction, the total area of sloping farmland reduction will reach 4436.5 km$^2$ by 2020, and the area reductions for 2–5°, 5–10°, 10–15°, 15–20°, 20–25°, and >25° sloping farmlands will be 1101.25, 1452.5, 721.3, 408.5, and 260,493 km$^2$, respectively (Figure 5).

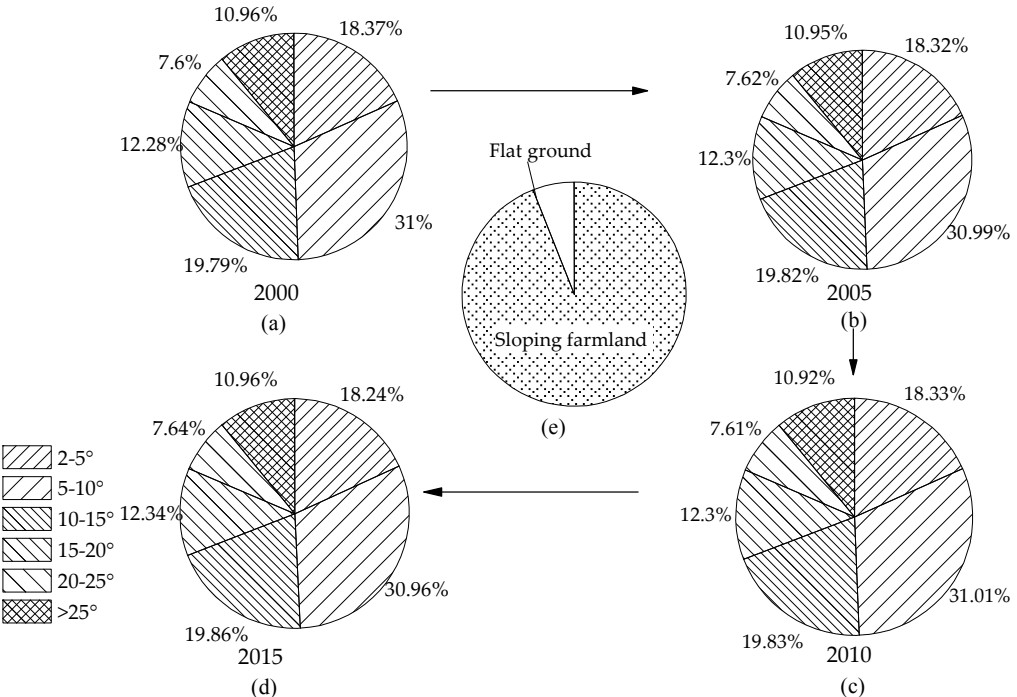

**Figure 4.** The proportion graph of the slope cultivated area with different slope grades. Note: (**a**–**d**) refer to the proportion graph of the slope cultivated area with different slope grades in 2000, 2005, 2010, and 2015, respectively. (**e**) Proportion of slope cultivated land and flat area in Sichuan Province.

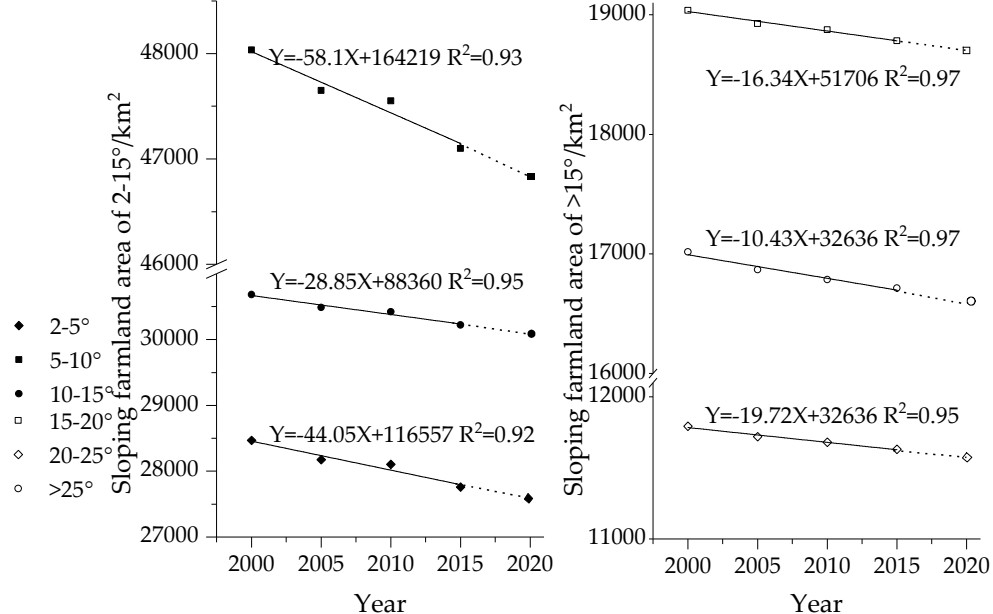

**Figure 5.** Linear regression diagram of the slope cultivated land area with time. Note: The solid line part is the actual change trend, while the dotted part is the forecast data based on the existing data.

### 3.3. Spatial Variation of Sloping Farmland Area

The total sloping farmland area of Leshan, Neijiang, Bazhong, Meishan, Suining, Mianyang, Guangyuan, the Aba Tibetan and Qiang Autonomous Prefecture, Deyang, Yibing, the Liangshan Yi Autonomous Prefecture, and Nanchong decreased by 100–200 km². The Ganzi Tibetan Autonomous Prefecture, Luzhou, Ya'an, Dazhou, Guang'an, Ziyang, Panzhihua, and Zigong decreased by less than 100 km², and Zigong City and Panzhihua City decreased by 54 and 49 km², respectively. From 2000 to 2015, the sloping farmland decreased by 691.2 km² in Chengdu, which was six times more than that of other cities.

The area of sloping farmland steeper than 25° severely decreased in western Sichuan province, especially in the three autonomous prefectures of ethnic minorities, where the total decreased area was up to 140 km² from 2000 to 2015. The gently sloping farmland decreased more than the steeply sloping farmland in eastern Sichuan Province, especially in Mianyang, Deyang, Nanchong, and Suining. The areas of sloping farmland between 2–5° and 5–10° in Chengdu City decreased by 230.2 km² and 305.1 km², respectively, between 2000 and 2015. Furthermore, the Nanchong, Mianyang, and Deyang areas of sloping farmland, which were less than 10°, decreased by 141, 136, and 138 km², respectively, which accounted for more than 70% of the total sloping farmland (Figure 6).

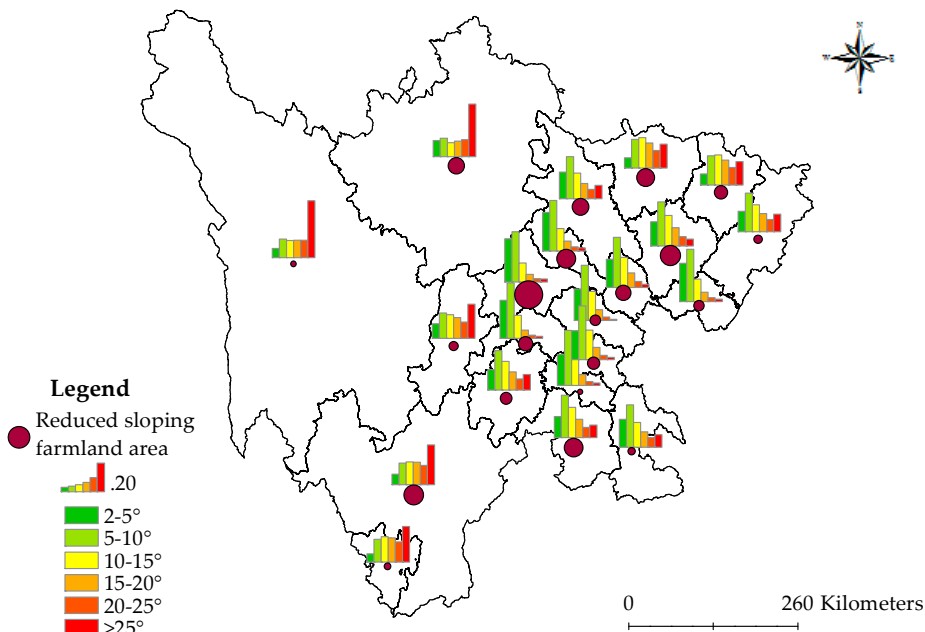

**Figure 6.** Sloping farmland area change in Sichuan province from 2000 to 2015. Note: As the legend shows, the total reducing area of slope cropland and the sloping farmland reduced area of each gradient were labeled using different legends.

### 3.4. Driving Force Analysis of the Changed Sloping Farmland Area

According to the variation characteristics of sloping farmland, in Sichuan province, the total area of sloping farmland maintained a general downward trend from 2000–2015. Therefore, indexes representing regional economic development, population, urbanization level, and agricultural intensification factors were selected to analyze the driving mechanism for the change of sloping farmland. The total population of Sichuan province doubled from 1952 to 2000, and the increasing trend slowed over the period from 2000–2015 (Figure 7). Urbanization was maintained at a 1% annual growth rate, while the output value of the tertiary industry and GDP per capita achieved sustained growth, though these growth rates significantly increased after 2000. The overall trend of the construction area increased with great fluctuations. The degree of intensity increased from 1990 onwards, whilst the sown area of crops showed little change. The output value of the tertiary

industry and the GDP per capita reflected the most significant increases, followed by the total mileage of highways and the urbanization rate. The total population, multiple cropping indexes, and the sown area of crops experienced slight changes. From 2000 to 2015, the output value of the tertiary industry, the GDP per capita, the construction area, and the total mileage of highways rose 7.47 times, 6.42 times, 3.54 times, and 3.47 times, respectively (Figure 8). The urbanization rate, multiple cropping index, total population, and the sown area of crops increased by 78.02%, 9.50%, 8.26%, and 0.84%, respectively.

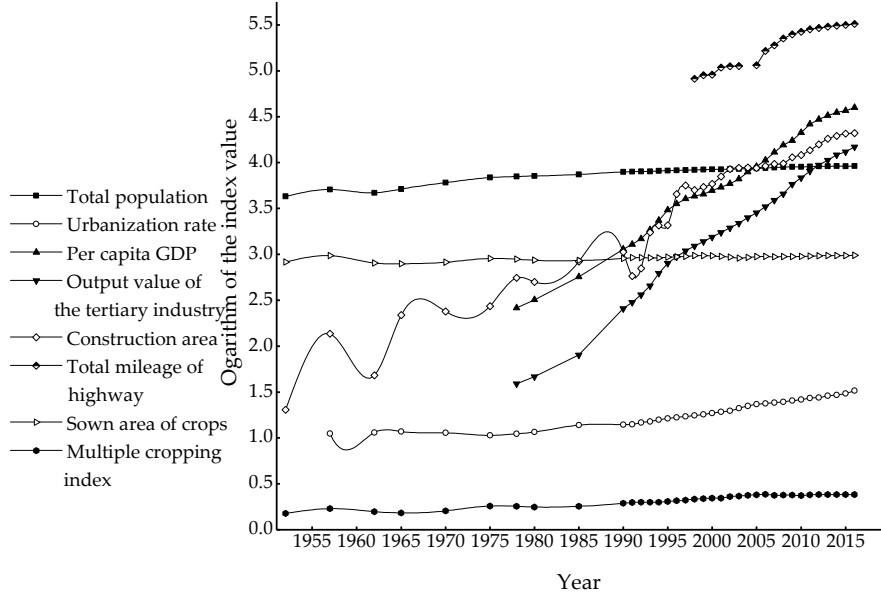

**Figure 7.** Driving force indexes of slope cropland over the calendar year in Sichuan province. Note: In order to gain a clearer understanding of the changing trends, all the indicators have been collated from the earliest years of statistics. Data on the total population, construction area, sown area of crops, and multiple cropping index are dated from 1952, the urbanization rate dates from 1957, the GDP per capita and output value of the tertiary industry are dated from 1978, and the total mileage of highways is dated from 1998.

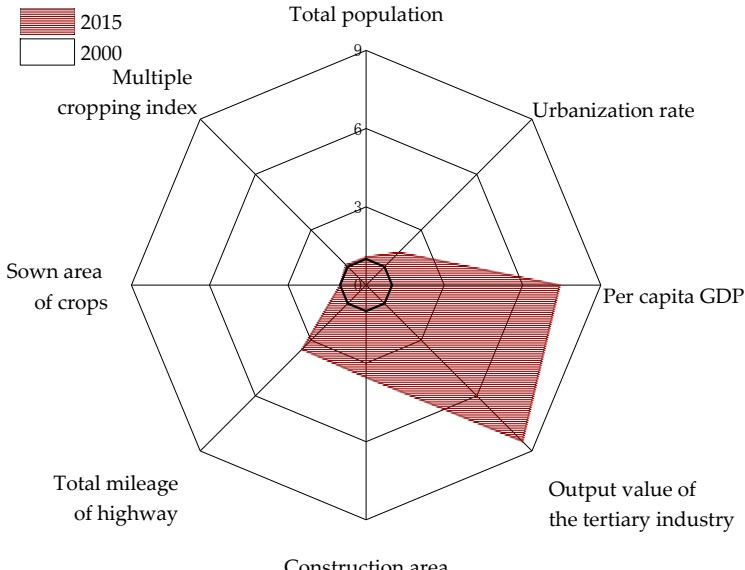

**Figure 8.** Driving forces of slope cropland in Sichuan province from 2000 to 2015. Note: Assuming that the indicator for the year 2000 was 1, this figure represents the magnitude ranges from 2000 to 2015.

3.4.1. Correlation Analysis between the Sloping Farmland Area and Its Driving Force

Slope cultivated land was strongly related to the multiple cropping index, with a positive coefficient of 0.698. However, the other factors were negatively correlated to sloping farmland, where the correlations ranged from strong to weak as follows: X5 > X3 > X4 > X1 > X2 > X6 > X7, and the corresponding correlation coefficients were −0.961, −0.949, −0.944, −0.937, −0.926, −0.898, and −0.301, respectively. Social and economic factors were negatively correlated to the sloping farmland, and most of the correlation coefficients were above −0.7. The correlation between agricultural intensity factors (sown area of crops, multiple cropping index) and sloping farmland area was less significant than that of the social and economic factors, and it showed obvious regional characteristics, being both positive and negative (Table 1).

**Table 1.** Correlation coefficient of the driving force index and slope cultivated land.

| | X1 | X2 | X3 | X4 | X5 | X6 | X7 | X8 |
|---|---|---|---|---|---|---|---|---|
| Sichuan Province | −0.937 | −0.926 | −0.949 | −0.944 | −0.961 | −0.898 | −0.301 | 0.698 |
| Chengdu | −0.973 | −0.995 | −0.949 | −0.899 | −0.921 | −0.886 | 1.000 | 0.78 |
| Zigong | −0.862 | −0.951 | −0.817 | −0.933 | −0.896 | −0.835 | −0.808 | −0.145 |
| Panzhihua | −0.797 | −0.874 | −0.998 | −0.996 | −0.952 | −0.913 | −0.99 | 0.116 |
| Luzhou | −0.777 | −0.975 | −0.954 | −0.963 | −0.994 | −0.815 | −0.889 | 0.273 |
| Deyang | −0.886 | −0.914 | −0.974 | −0.941 | −0.905 | −0.82 | 0.476 | 0.082 |
| Mianyang | −0.959 | −0.895 | −0.948 | −0.931 | −0.909 | −0.908 | 0.119 | −0.051 |
| Guangyuan | −0.261 | −0.969 | −0.091 | −0.984 | −0.991 | −0.898 | −0.988 | 0.173 |
| Leshan | −0.781 | −0.929 | −0.945 | −0.972 | −0.989 | −0.874 | 0.812 | 0.161 |
| Suining | −0.108 | −0.903 | −0.933 | −0.961 | −0.978 | −0.825 | 0.41 | 0.088 |
| Neijiang | −0.835 | −0.795 | −0.881 | −0.844 | −0.944 | −0.836 | −0.019 | −0.313 |
| Nanchongngng | −0.769 | −0.895 | −0.895 | −0.896 | −0.922 | −0.827 | −0.797 | −0.091 |
| Meishan | −0.8 | −0.982 | −0.969 | −0.993 | −0.994 | −0.85 | −0.218 | 0.279 |
| Yibin | −0.944 | −0.895 | −0.947 | −0.96 | −0.964 | −0.904 | −0.999 | 0.003 |
| Guang'an | −0.732 | −0.994 | −0.968 | −0.971 | −0.977 | −0.727 | −0.954 | 0.288 |
| Dazhou | −0.805 | −0.971 | −0.985 | −0.996 | −0.99 | −0.85 | −0.809 | 0.287 |
| Ya'an | −0.652 | −0.993 | −0.974 | −0.99 | −0.974 | −0.16 | 0.661 | 0.42 |
| Bazhong | −0.71 | −0.966 | −0.974 | −0.981 | −0.993 | −0.89 | −0.982 | 0.145 |
| Ziyang | −0.771 | −0.766 | −0.845 | −0.843 | −0.79 | −0.803 | −0.063 | −0.207 |
| Aba | −0.888 | −0.639 | −0.657 | −0.797 | −0.785 | −0.829 | 0.283 | −0.422 |
| Ganzi | −0.791 | −0.476 | −0.684 | −0.761 | −0.38 | −0.785 | 0.184 | −0.772 |
| Liangshan | −0.994 | −0.774 | −0.984 | −0.98 | −0.998 | −0.998 | −0.992 | −0.151 |

Note: social factors (total population/10 000 people (X1), urbanization rate/% (X2)); economic factors (per capita GDP/yuan (X3), the output value of the tertiary industry/10,000 yuan (X4), total mileage of highways/km (X5), construction area/km$^2$ (X6)); and agricultural intensity factors (sown area of crops/km$^2$ (X7) and the multiple-cropping index (X8)).

3.4.2. Regional Characteristics of the Driving Force of Sloping Farmland Area Change

There were two principal components with eigenvalues that were greater than 1. The cumulative contribution rates of the first two principal components were all above 90%, except for Suining with 87%. This fully satisfied the requirements of the principal component analysis in the regional scope, and could express the comprehensive information of the driving force indexes of sloping farmland. The eigenvalue of the first principal component was much higher than that of the second principal component. The eigenvalues of the first factor were all above 5, except for Ya'an and Guangyuan, with values of 4.955 and 4.789, respectively. The eigenvalue of the first principal component was 7.573 in Chengdu, which was the highest. The eigenvalues of the second principal component were all greater than 1, except for Chengdu, which had an eigenvalue of 0.302, and its contribution rate was between 10% and 30%. The second principal component also had a certain driving effect on the change of sloping farmland area (Figure 9).

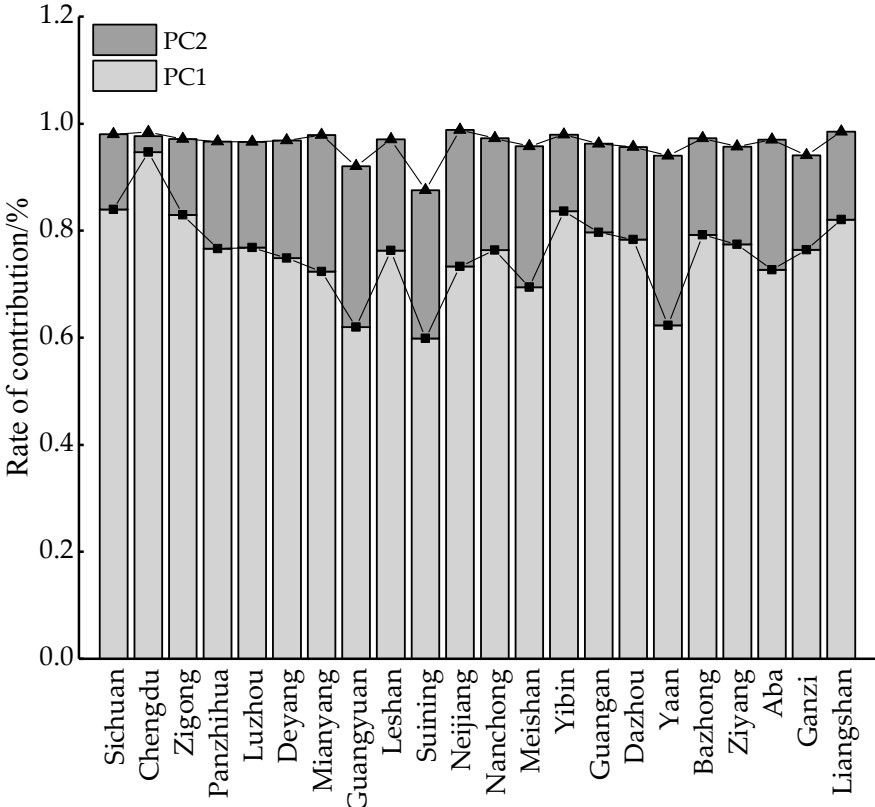

**Figure 9.** The first and second principal component contribution rates in Sichuan province.

The first principal component mainly reflected the social and economic factors, and the function of Sichuan province was Z = −0.977X2 + 0.992X3 + 0.998X4 + 0.993X5 + 0.925X6, which was closely related to economic factors (X3, X4, X5, X6). The loading sequence was: output value of the tertiary industry X4, total mileage of highways X5, per capita GDP X3, and construction area X6. At the same time, the urbanization rate X2 also had a high correlation with the change of sloping farmland area in Sichuan province. The first driving force of 21 cities (prefectures) of Sichuan province involved seven driving forces, where the most important ones were the four economic factors (X2, X3, X4, and X5) and the urbanization rate (X6). The total population and sown area of crops had a significant impact on the change of sloping farmland area in a few regions. The degree of influence of the driving force load indexes of different cities (prefectures) had great regional differences (Table 2).

According to the loads of the different driving factors on the sloping farmland, 22 cities (prefectures) in Sichuan province could be divided into the following two types: the social–economic type and the social–economic–agricultural intensive type. The social–economic type referred to the areas where the change of sloping farmland area was significantly correlated with the social and economic factors, which were mainly distributed in the Chengdu Plain and its surrounding regions, as well as the Aba and Ganzi autonomous prefectures. Meanwhile, it could also be subdivided into two types, which were affected and not affected by the total population X1. Seven cities (prefectures) were affected by the total population X1, namely, Chengdu, Deyang, Neijiang, Meishan, Ziyang, and the Aba and Ganzi autonomous prefectures. The seven other cities were mainly affected by the urbanization process and economic development. The social–economic–agricultural intensive type referred to the areas where the change of sloping farmland was significantly correlated to social factors, economic factors, and agricultural intensification factors. These cities (prefectures) were mainly distributed at the edge of southeastern Sichuan province. The group included seven cities (prefectures), such as Zigong, Panzhihua, and Yibin.

**Table 2.** Matrix of the first principal component after rotation.

| Region | X1 | X2 | X3 | X4 | X5 | X6 | X7 | X8 |
|---|---|---|---|---|---|---|---|---|
| Sichuan Province | | 0.977 | 0.992 | 0.998 | 0.993 | 0.925 | | |
| Chengdu | 0.993 | 0.978 | 1.000 | 0.989 | 0.994 | 0.950 | | |
| Zigong | 0.934 | 0.959 | 0.987 | 0.992 | 0.983 | 0.945 | 0.989 | |
| Panzhihua | | | 0.998 | 0.993 | 0.952 | 0.926 | 0.990 | |
| Luzhou | | 0.958 | 0.994 | 0.997 | 0.995 | | | |
| Deyang | 0.961 | 0.915 | 0.982 | 0.956 | 0.998 | 0.972 | | |
| Mianyang | | 0.965 | 0.997 | 0.999 | 0.999 | 0.970 | | |
| Guangyuan | | 0.935 | | 0.999 | 0.993 | 0.973 | | |
| Leshan | | 0.971 | 0.991 | 0.999 | 0.977 | | | |
| Suining | | 0.939 | 0.990 | 0.999 | 0.994 | | | |
| Neijiang | 0.922 | 0.969 | | 0.995 | 0.967 | 0.993 | | |
| Nanchong | | 0.960 | 0.999 | 0.999 | 0.997 | 0.930 | | |
| Meishan | 0.904 | 0.958 | 0.999 | 0.992 | 0.952 | 0.937 | | |
| Yibin | 0.969 | 0.931 | 0.997 | 0.999 | 0.999 | 0.976 | 0.952 | |
| Guang'an | | 0.966 | 0.999 | 0.998 | 0.975 | | 0.992 | |
| Dazhou | 0.907 | 0.898 | 0.994 | 0.989 | 0.993 | 0.942 | 0.891 | |
| Ya'an | | 0.972 | 0.987 | 0.998 | 0.938 | | | |
| Bazhong | | 0.934 | 0.996 | 1.000 | 0.967 | 0.959 | 0.998 | |
| Ziyang | 0.911 | 0.955 | 0.985 | 0.985 | 0.992 | 0.934 | | |
| Aba | 0.917 | 0.959 | 0.980 | 0.980 | 0.958 | 0.934 | | |
| Ganzi | 0.967 | | 0.994 | 0.982 | | 0.987 | | |
| Liangshan | 0.988 | | 0.996 | 0.994 | 0.996 | 0.988 | 0.977 | |

The second principal component was mainly reflected in the agricultural intensification factors. Analysis of regional differences in the driving forces of the second principal component showed that the eigenvalue of the second principal component in Chengdu was less than 1, and the contribution rate was only 3.02%; thus, the second principal component in Chengdu was excluded. However, the contribution rates of the other cities (prefectures) were about 10% to 30%; therefore, the analysis was necessary. In the second principal component, the total population X1 of Sichuan province had the highest load coefficient, indicating that population growth still had a significant impact on the change of sloping farmland in Sichuan province. Except for Deyang, Guangyuan, Suining, Ya'an, and the Ganzi Tibetan Autonomous Prefecture, the 16 other cities (prefectures) showed significant correlations with the multiple cropping index X8 in the second principal component, which indicated that the change in the multiple cropping index would cause a change of sloping farmland. However, its influence was lower than that of the social and economic factors on sloping farmland (Table 3).

**Table 3.** Matrix of the second principal component after rotation.

| Region | Index | Load Coefficient | | Index | Load Coefficient |
|---|---|---|---|---|---|
| Sichuan Province | X1 | 0.473 | Nanchong | X8 | 0.939 |
| Chengdu | - | - | Meishan | X8 | 0.974 |
| Zigong | X8 | 0.97 | Yibin | X8 | 0.982 |
| Panzhihua | X8 | 0.965 | Guang'an | X8 | 0.979 |
| Luzhou | X8 | 0.898 | Dazhou | X8 | 0.99 |
| Deyang | X7 | 0.77 | Ya'an | X6 | 0.852 |
| Mianyang | X8 | 0.952 | Bazhong | X8 | 0.964 |
| Guangyuan | X1, X3 | 0.906,0909 | Ziyang | X8 | 0.921 |
| Leshan | X8 | 0.866 | Aba | X8 | 0.941 |
| Suining | X1 | 0.987 | Ganzi | X7 | 0.771 |
| Neijiang | X8 | 0.968 | Liangshan | X8 | 0.987 |

## 4. Discussion

Sichuan is a typical sloping farmland province, where the sustainable utilization of sloping farmland resources is of great practical significance to the Sichuan province. Slope cultivated land ≤15°, where many high-yielding places are located, was the basic component of cultivated land and it is an important guarantee to maintaining agricultural production in the Sichuan province. Alternately, 15–25° slope cropland has generally declined in quality. When using such arable land, it is necessary to support the use of biological measures, engineering measures, and other water conservation measures to improve the sustainable production capacity of sloping farmland [48]. Slope >25° cultivated land was unsuitable for farming. The study indicated that the rate of reduction of the gentle slope area (especially less than 10°) was faster than others, while the slope cultivated land area of 20–25° reduced at the slowest rate. The invisible increases in the utilization pressure of moderately steep slope cultivated land then put forward higher requirements of the farming technology of steep slope cultivated land, such as terrace and farming techniques [49], to ensure the sustainable development of sloping farmland and food security.

The strongest factors with effects on sloping farmland were economic factors, followed by social factors, and then agricultural development factors were the least. Both social and economic factors had a significant negative correlation with the area of sloping farmland, and a greater impact on the area of sloping farmland was found with an increase in these factors. Many studies have shown that the change of cultivated land is closely related to the social economy, agricultural development, population pressure, agricultural structure adjustment, and the urbanization level [50]. Owing to the influence of comparative economic utilization, the sloping farmland would be transferred from the low-output industries to the high-output industries [51]. With rapid economic growth, advancement of urbanization, population growth, and agricultural development, the area of sloping farmland will inevitably change. The increase in social factors like population pressure and urbanization rate inevitably caused the increase of construction land, where construction land generally occupied high-quality gently sloping farmland [52]. Since the implementation of market oriented economic reforms in 1978, China has been on a track of rapid urbanization. The unprecedented urbanization in China has resulted in substantial cultivated land loss and the rapid expansion of urban areas. Moreover, there are regional differences in the influence of the driving forces of slope cultivated land, wherein the areas of gently sloping farmland <10° severely decreased in areas experiencing rapid economic development and high urbanization rates, such as Chengdu. Ecosystem services also affected the slope cultivated land [53], such as agricultural intensification and ecological restoration. The influence of agricultural intensification factors on the change of sloping farmland area was relatively weak. Ecological restoration played a certain effect, where the farmland area with a >25° slope sizably decreased, especially in the western region of Sichuan province, as well as in the three major minority autonomous prefectures. China's "Grain To Green" program has been widely praised as the world's largest and most successful payment for ecosystem services program [54]. However, we need to continue to implement and promote the program, because there is still a lot of area to cover. Based on the regularity of regional differences, we should adapt to local conditions and rationally allocate the utilization of sloping farmland, we should try to synchronize the quality of sloping farmland with economic growth and population increase, and we should improve the utilization technology of sloping farmland to relieve the current decreasing trend of sloping farmland.

## 5. Conclusions

From 2000 to 2015, farmland has reduced from 25.1% to 24.6% of the entire land in the Sichuan province. Sloping farmland area has decreased to 3263 km$^2$, and the reduction rate of gentle slope cultivated land was faster than that of moderately steep sloped land, while the reduction degree of slope cultivated land in developed areas was greater than that in underdeveloped areas. Meanwhile, the farmland area with a >25° slope, which is not suitable for farming, has decreased significantly, especially in the western region and in the three major minority autonomous prefectures.

Social factors and economic factors had a significant negative correlation with the area of sloping farmland in the Sichuan Province, while the load coefficient of agricultural intensity factors was relatively small. From the comprehensive inference of the driving forces, the strongest factors with effects on sloping farmland were economic factors, followed by social factors, and then agricultural development factors were the weakest.

The first principal component mainly reflected the social and economic factors with contribution rates of 60%–90%. In this section, 21 cities (prefectures) in the Sichuan Province could be divided into the social–economic driving type (14 cities and prefectures), and also the social–economic–agricultural intensification driving type (7 cities and prefectures). The second principal component, with a contribution rate of 10% to 30%, mainly reflected the impact of agricultural intensity factors, especially the multiple cropping index on the sloping farmland.

In summary, development in Sichuan was unbalanced, which was reflected by faster development of the economy and society compared with agricultural development. As a result, a lot of high-quality cultivated land on gentle slopes was occupied, increasing the pressure on the use of sloping farmland. To ensure the sustainable development of sloping farmland and food security, we should strengthen the protection of sloping farmland within the scope of social and economic development.

**Author Contributions:** Conceptualization, Q.Z. and L.Q.; methodology, Q.Z. and Y.D.; software, M.X.; validation, L.Q. and M.X.; formal analysis, M.X. and L.Q.; investigation, H.A.H. and L.Z.; data curation, Q.Z. and M.X.; writing—original draft preparation, M.X.; writing—review and editing, Q.Z., L.Q., H.A.H., and M.X.; project administration, Q.Z.; funding acquisition, Q.Z.

**Funding:** This research was supported by the Special Fund for Scientific Research on Public Causes (201503119). Project was also supported by the National Natural Science Foundation of China (41501302); the National Science and Technology Major Project of the Ministry of Science and Technology of China (2015ZX07203-007), and the Research and Development Support Program from the China Institute of Water Resources and Hydropower Research (SC0145B372017).

**Conflicts of Interest:** The authors declare no conflict of interest.

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
