# Peer review of "Spatiotemporal Changes and the Driving Forces of Sloping Farmland Areas in the Sichuan Region"

_sustainability, doi:10.3390/su11030906_

Round 1
Reviewer 1 Report
The paper entitled “Spatiotemporal Changes and Driving Forces of Sloping Farmland Areas in Sichuan Region Based on the GIS Image” presents and in-depth assessment of the principal land use variations happened in the Sichuan Region on a certain terrain slope and see how this changes are influenced by socio-economic factor.
The topic is of great interest since this kind of assessment reveals with a better detail the driving forces of a certain land use changes on specific sites and therefore it is much more appreciable in the way this kind of study suggest specific policies to mitigate or control land use changes by different measures. Thus the paper is of interest for the scientific community and also the approach followed by authors seems to be scientifically grounded.
The main problem of this manuscript is its confusion in general. Meaning that the reader see many repetition in the content, some phrases are too vague or anticipate something that is not clearly presented in the methodology and , least, the authors did not address properly the main question of the methodological approach: the Land Use Change (LUC) analysis.
A study like this is entirely based on LUC but the paper does not introduce (nor refer to) the different methodologies of LUC (the differential rather than the cross-tabulation matrix), how to assess by GIS mapping the change, which functions are important do map its distribution, and did not explain if the threshold of land use dataset used are effectively comparable. I suggest to authors to read some works of Peter Verbourg or Veldkamp (see at least these works)
Veldkamp, A. and Verburg, P.H. (2004), “Modelling land use change and environmental impact”, Journal of Environmental Management, Vol. 72 Nos 1-2, pp. 1-3.
Verburg, P.H., Schot, P.P., Dijst, M.J. and Veldkamp, A. (2004), “Land use change modelling: current practice and research priorities”, GeoJournal, Vol. 61 No. 4, pp. 309-324.
I made a synthesis of their approaches in this work (not to cite but to help you to approach at LUC analysis in the correct way).
Stefano Salata, (2017) "Land use change analysis in the urban region of Milan", Management of
Environmental Quality: An International Journal, Vol. 28 Issue: 6, pp.879-901, https://doi.org/10.1108/
MEQ-07-2016-0049
I have also to state that there is another structural problem: this kind of assessment normally considers the ecosystem service assessment as the main object of analysis since ESs better explain the driving forces of LUC rather than other variables. But in the paper ES are not introduced nor considered (maybe you have to explain why…)
For what concerns the detailed comments:
Line 32 See you references in the text (wrong style)
Line 79 it in not clear in the introduction which methodology you used to compare land use change variations
Line 92 of in
Line 95-98 rephrase (unclear) “based on”???
Lines 114-115 repetition (already introduced)
Line 117 space after PCA
Line 153-154 what does it means?
Figure 2 instead of the land use indexes it is better to show the variation. And also: why it is expected that slopes change across the time? The slope???
Lines 160-164 how did you calculate this?
Lines 170-172 why prediction? Which method? Regression?
Lines 201-204 now you re-introduce LUC
Lines 208-213 third time you write this…
Lines 221-222 spacing
Lines 237-238 ???
Line 246 spacing
Table 1 what year? Is the variation?
Lines 275-277 not proper to put a note in that way
Line 289 spacing
Author Response
Dear reviewer:
Thank you for your attention, I upload the response as a Word file.

Reviewer 2 Report
Please see my attached comments. Thanks.

Author Response

(The authors gave the same response as above.)

Reviewer 3 Report
The presented manuscript concerns important issue of Spatiotemporal Changes and Driving Forces of Sloping Farmland Areas in Sichuan Region Based on the GIS Image. What is the significance level assumed in the calculation of the coefficient of determination R2. Give the legend on the right side of the Figures, eg. in Figure 5. Driving force indexes of slope cropland over the calendar year of Sichuan province. Why You used in the analysis 5 years period of observation? (eg. 2000, 2005, 2010 and 2015), what about the recent years 2016 and 2017? Abstract should be more concise. Include in the introduction and the conclusion the value added with respect to existing research. Also the introduction should include information about problem, which should be investigated, background and state-of art that explains the problem, as well as reasons for conducting the research. Authors need to rewrite the hypothesis/purpose of the work more clearly. Add references to equations. The manuscript should be prepared according to journal guidelines, and should be really carefully edited (unnecessary spaces, etc.). Authors do not explain well, where is the novelty of the distinguished method. The section should be more focused and based on the obtained results. Conclusions are not sufficiently described. How in practice the results of the presented work can be used? Some information about practical use of the obtained results, both in the section Results and Conclusions should be underlined. Point 2.3. add some information about limitation of using Principal component analysis). Analysis Much more consistency needs to be achieved in the interplay between results and conclusions, and it needs to be strongly revised.
Author Response

(The authors gave the same response as above.)

Round 2
Reviewer 1 Report
It’s a bit displeasing to see this work after the first major revision finding out similar weaknesses to the first draft.
Sorry to be so insistent but main problems remains the same.
1. English. The paper it’s written in a bad English and a bad stylistic sentencing;
2. Structure. It’s confused and the discussion is based on a method that didn’t explain how you reach indicators and map. As written before Land Use Change assessment has to be adequately introduced and to make a good LUC assessment can condition your entire work.
3. Unclear. You used a lot of analysis and methodologies but you didn0t adequately introduced in you work: LUC, regression, PCA… all this can be an object for a single scientific work. Why don’t cut your manuscript around just one analytical method and make simple-and-clear considerations?
4. Rush. Please re-read your text. It’s full of mistakes. Some phrases are unreadable. I had the impression that you rushed to your revision but without taking the adequate time to do it. But this will affect your further work for this project of publication.
All specific comments in the attached file.
Best.

Author Response
Dear reviewer:
Thank you for your attention, I upload the response as a PDF file.

Reviewer 2 Report
Thanks authors to address my concerns. Other than some grammar issue, I do not have any other questions.
Author Response
Dear reviewer:
Thank you for your good advice. We have checked the grammar issue, and used MDPI English editing service to improve our expressions.

Reviewer 3 Report
Please present the results and conclusions in more clearly way.
Author Response
Dear reviewer:
Thank you for your attention. We have re-read and revised the results and conclusions to make it more clearly. And the revision is tracked in article.

Round 3
Reviewer 1 Report
I see the main corrections in the manuscript and I found that the stylistic problem (English editing) has been addressed now. The structure has been slightly revised accordingly.
I still found this manuscript redundant of a redundant methodological process which can be analyse throughout a detailed LUC assessment. Nonetheless now I think the manuscript reached a reasonable quality to be published.
Please, re-read carefully the text. There are still some mistakes…
Line 66 space
Line 85 “S”
Line 102 sloping farmland area is a title? Line 136 the same
Line 157 space
Line 182 space
Line 195 spacing
Author Response
Dear reviewer:
Thank you for your comments, suggestions and references, land use change models are tools for understanding and explaining the causes and consequences of land use dynamics. In this article, we use another method to analyze LUC. We’ll study LUC assessment more deeply for further studies in this program. And we have checked the whole text carefully. The details are as follows:
Point 1: Line 66 space
Response 1: We have added space in line 66.
Point 2: Line 85 “S”
Response 2: We have changed “s” to “S” in line 85.
Point 3: Line 102 sloping farmland area is a title? Line 136 the same
Response 3: We have set this sentence as a title in line 102 and 136, revised as follows:
2.2.1. Data of sloping farmland area
2.2.2. Data of driving forces.
2.3.1. Spatiotemporal changes analysis of sloping farmland.
2.3.2. Driving force analysis.
Point 4: Line 157 space
Response 4: We have added the space in line 157.
Point 5: Line 182 space
Response 5: We have added the space in line 182.
Point 6: Line 195 spacing
Response 6: We have deleted the space in line 195.
